# Automatic Identification of a Depressive State in Primary Care

**DOI:** 10.3390/healthcare10122347

**Published:** 2022-11-22

**Authors:** Xiaoqian Liu, Xiaoyang Wang

**Affiliations:** 1Institute of Psychology, Chinese Academy of Sciences, Beijing 100107, China; 2Department of Psychology, University of Chinese Academy of Sciences, Beijing 101408, China

**Keywords:** depression, CES-D, facial movements, primary care, social media

## Abstract

The Center for Epidemiologic Studies Depression Scale (CES-D) performs well in screening depression in primary care. However, people are looking for alternatives because it screens for too many items. With the popularity of social media platforms, facial movement can be recorded ecologically. Considering that there are nonverbal behaviors, including facial movement, associated with a depressive state, this study aims to establish an automatic depression recognition model to be easily used in primary healthcare. We integrated facial activities and gaze behaviors to establish a machine learning algorithm (Kernal Ridge Regression, KRR). We compared different algorithms and different features to achieve the best model. The results showed that the prediction effect of facial and gaze features was higher than that of only facial features. In all of the models we tried, the ridge model with a periodic kernel showed the best performance. The model showed a mutual fund R-squared (R2) value of 0.43 and a Pearson correlation coefficient (r) value of 0.69 (*p* < 0.001). Then, the most relevant variables (e.g., gaze directions and facial action units) were revealed in the present study.

## 1. Introduction

Mental health is an important and growing public health concern among the general population. Considering that as time goes by, the severity of mild mental health problems may increase if they are not given timely attention or help [1], people try to integrate mental health services within primary care to provide extensive and timely screening of people’s mental states [2,3].

The Center for Epidemiologic Studies Depression Scale (CES-D) [4] is a commonly used depression screening instrument to identify depression among the general population [5]. The psychometric characteristics of the CES-D scale appear to be relatively constant across different groups [6,7]. Additionally, its effectiveness in primary care scenes has been proven many times [8,9]. Although the CES-D is a good match for depression screening in primary care, people usually use the PHQ-9 instead because the CES-D has too many items to screen for [10,11]. However, given the high sensitivity of CES-D [11,12], which is more important than specificity in primary care, a simple, but effective CES-D automatic identification system can be helpful.

People’s nonverbal behavior is related to their psychological state, and psychiatrists have always combined the patient’s nonverbal behavior to make a diagnosis [13,14,15]. Then, aided by emerging technology, researchers quantify nonverbal behavior and explore the relationship between nonverbal behavior and a depressive state [16], the therapeutic effect of depression [17], and the development of depression [18]. Recently, with the development of artificial intelligence technology, machine learning methods have been applied to a wider range of research fields [19,20]. An automatic recognition model has been proposed to realize the automatic recognition of depression through nonverbal behavior (such as by voice [21], face, etc.). Some researchers proposed a deep regression network of DepresNet to learn the depression representation with visual interpretation and realized the depression detection method based on facial image analysis [22,23]. In Ref. [24], researchers effectively recognized depression by fusing facial and voice data. Meanwhile, in Ref. [25], researchers used 3D facial expressions and spoken language to recognize depression based on the machine learning method.

Facial expression and gaze are thought to be associated with depression. Studies found that a depression group opens their eyes smaller, have a longer duration of blinking [26], and have less eye contact with others [27] compared with a general group or a cured group. Other studies used facial activities to effectively predict people’s depression levels [28,29,30]. Moreover, the integration of different features brings a better prediction effect than a single feature [31]. Considering that both facial activities and gaze are visual data that are easily accessible, using facial activities and gaze data together to predict CES-D scores may be a simple, yet effective method.

In summary, this study integrated facial activities and gaze data to establish an automatic recognition model of CES-D scores. We assume that the prediction model can predict CES-D scores effectively to indicate depressive states, and the model fusing facial activities and gaze is better than the model containing only facial activities. This study can be applied to primary care to simply identify depressive states.

In Section 2, we introduce the process of data acquisition, facial data acquisition, facial feature extraction methods, and model-building methods. In Section 3, the performance evaluation results of the depression recognition model are reported. The research results are analyzed and discussed in Section 4.

## 2. Method

### 2.1. Participants and Procedure

A total of 152 students and workers from the University of Chinese Academy of Sciences participated in this study, of which 79 were male and 73 were female. To participate in the study, participants had to meet three inclusion criteria: (1) participants needed to be at least 18 years old; (2) participants needed to be fluent in Mandarin; and (3) participants needed to be physically healthy and able to make normal facial activities.

### 2.2. Measures

To build a predictive model between facial expression videos and CES-D scores, this study designed a facial data collection experiment with a self-introductory situation. The self-introductory task made the subjects show more attitude toward themselves [32], and this is also common in primary healthcare. The data acquisition experiment was carried out according to the following steps in Figure 1: (1) First, the demographic information of each subject was recorded. (2) Then, the participants were asked to complete the CES-D scale. (3) During the experimental task (the self-introduction), the participants were recorded introducing themselves for one to three minutes using a high-definition camera (the frame rate was 30 Hz). The distance between the camera and the participant was controlled to be 3 m to exclude the influence of distance on the intensity of any facial movements. Meanwhile, participants were asked to stand in a bright place without direct sunlight to prevent overexposure or dark light.

### 2.3. Instruments

This study used the Center for Epidemiologic Studies Depression Scale (CES-D) [4] as a tool to identify depression among the general population. The scale contains 20 questions in total. Each question describes a possible or recent feeling, such as: “In the past week, I have been bothered by something that doesn’t bother me normally”, “In the past week, I don’t want to eat, and my appetite is poor”, etc. The score ranges from 20 to 80. The items primarily measure the affective and somatic aspects of depression. According to the criteria proposed by the original author, depression exists in individuals with CES-D scores of 36 or more [4].

OpenFace [33] is a state-of-the-art tool for analyzing human behavioral video data, and it can recognize the movements of 17 facial action units [34], as well as estimate 8 indices of eye gaze [35] for each video frame. The specific meaning of each visual feature is provided in Table 1.

### 2.4. Data Preprocessing

After obtaining the full-body videos, we first cut the videos to 500 px × 500 px facial videos containing the subjects’ whole face. Then, these facial videos were grayed to reduce irrelevant information, such as color.

### 2.5. Feature Extraction and Reduction

Seventeen facial features and eight gaze features for each frame were extracted from grayed facial videos using OpenFace. Then, we intercepted each feature data point from frame 300 to frame 2100, which took approximately 1 min. The purpose of only intercepting the middle data was to eliminate the preparation time before and after the self-introduction. After that, the “all features” pattern of tsfresh 0.18.0 [36] was used to extract 787 time-series characteristics from each facial expression and gaze. Thus far, we have obtained a feature file; the number of rows is the number of subjects (152), and the number of columns is the number of time-series characteristics (25 × 787). Finally, normalization was conducted to balance the range of characteristics.

To prevent overfitting, we had to select dozens of features from this great quantity of features, which was related to the sample size. We carried out feature reduction and feature selection. For feature reduction, we used principal component analysis (PCA) to extract 8 principal components from 787 time-series characteristics of each feature. Compared with reduction from the time-series characteristics of all features, extracting principal components separately helps to ensure that all facial or gaze features can be preserved after unsupervised reduction. For feature selection, we calculated the F values between 200 principal components (25 × 8) and CES-D scores, and then we selected the principal components with the largest F value. The F value is the importance of the correlation coefficients between principal components and CES-D scores, and the calculation process is shown in Formula 1. Because of the uncertainty of how many principal components could achieve the best effect, we output the top (20, 25, 30, 35, 40, 45, 50, 55, 60) principal components.
(1)$$F=\frac{r^{2}}{\left(1-r^{2}\right)}\times df$$

In Formula 1, *r* represents the Pearson correlation coefficients between CES-D scores and values of one principal component. *df* represents the degrees of freedom. The larger the F value is more relevant than the principal component is to CES-D scores.

#### 2.5.1. Statistical Analysis

A regression model attached to L2 regularization and a kernel function was implemented in Python 3.7.6 [37], which was called kernel ridge regression (KRR). KRR is used for supervised learning problems where we use the multiple training features X_i (the selected principal components) to predict a target variable Y_i (the CES-D scores). Compared with other regression models, KRR has both the advantages of L2-norm regularization and the kernel function. L2-norm regularization can learn from all of the features and reduce the influence of outliers on increasing parameters to prevent overfitting [38]. Meanwhile, the kernel function can make the nonlinear relation map linear in a high-dimensional space for the principal components that are nonlinear with CES-D scores to resolve. Then, we compare the performance of KRR with linear ridge regression and kernel support vector regression (SVR).

#### 2.5.2. Model Testing and Validation

The KRR model can utilize a 5-fold cross-validation method to split the “training set” and the “testing set”. This method can balance the false high or false low performance caused by sampling deviation. Validation of the algorithm in predicting CES-D scores was calculated using the mean absolute error (MAE), mean squared error (MSE), R squared (R2), and Pearson correlation coefficient (r). In this study, Mae and MSE were related to the difference between the true scores and the predicted scores. Thus, the smaller indices showed a better performance. R2 reflects the interpretation degree of the predicted scores to the true scores, and a larger R2 represents a better model. R is the correlation coefficient between the true scores and the predicted scores; thus, a larger correlation coefficient is expected.

## 3. Results

A total of 79 males (51.97%) and 73 females (48.03%) were involved in this study. Their average CES-D score was 31.44, and the standard deviation was 7.05. According to the criteria (CES-D score ≥ 36) proposed by the original author [4], 37 of them (24%) had depression.

The study developed a KRR model to simulate the association between facial expression, gaze, and depression. After adjusting the parameters, the Pearson correlation coefficient of the best-performing algorithm was 0.69 (MAE 4.73, MSE 33.62, R2 0.43), which represented a strong correlation between the predicted values and the true scores. The trend of evaluation scores (r, MAE, MSE, R2) changing with the number of features is shown in Figure 2. We found that the model performed best when the 40 most relevant principal components were input.

To obtain a better prediction model and make it more accurate for primary care, we compared two common algorithms attached with different kernel functions (Table 2). The results show that the ridge regression model with the periodic kernel function is the best.

Eyes and faces are generally considered to be important areas of depression expression. However, this study also compared depression recognition performance with face-only features and facial and gaze features. The results showed that the combination of facial expression and gaze was better for predicting depression in all four evaluation indicators (Table 3).

In the best model, the 40 most relevant principal components were used. The model may have been more explicable if we knew which features these principal components belonged to and which features were more relevant to the CES-D scores. Thus, we investigated the original features of these 40 principal components. Then, we classified these 40 principal components into their features and added their F values to obtain the correlation degree between these features and the CES-D scores. We rank these effective features by correlation degree, as shown in Figure 3.

## 4. Discussion

In this study, we tried a machine learning approach to identify depressive states that may be used in primary healthcare. The results proved our hypothesis after we found that using video data can effectively predict depressive states. In addition, the combination of facial expression and gaze is better than only facial features in screening for depression. Furthermore, we investigated the correlation between these features and the CES-D scores.

We collected CES-D scores to be predicted based on video data. Then, we used Mae, MSE, R2, and r to compare the performance of different algorithms, different categories of features, and different numbers of principal components. We found that the KRR model with 40-dimensional principal components had the best performance, which was inputted with both facial and gaze features. Compared with some previous studies on auto recognition of depression, the proposed model has a smaller Mae value [39,40] and a larger *p*-value [41]. This shows that the model performed well.

Consistent with previous studies on the nonverbal expression of depression [13,16], this study found that both facial activities and gaze were associated with a depressive state. Among facial activities, AU20 (lip stretcher) was found to be the most relevant to the CES-D scores. One potential explanation is that the articulation of depressed people is different from that of the general population [42], and articulation is related to lip stretching [43]. In addition, we noted that AU9 (Nose Wrinkler) and AU10 (Upper Lip Raiser) were also closely related to CES-D scores, which are also representative of facial activities of disgust [44]. Considering the self-introduction tasks, this is in line with the fact that depressed and self-disgust feelings emerged as closely co-occurring and consistent symptoms (≥80% of depressed patients) [32]. Among gaze activity, eye gaze direction vectors were relevant to the CES-D scores. Such a relationship was also found in the correlation between eye gaze direction vectors and schizophrenic symptom severity [45].

The results also showed that the predictive effect of spliced facial and gaze features was better than that of facial features alone. When individuals are depressed, their eyes are not fully opened, which may indicate fatigue or decreased interest [26], and their faces may express negative emotions. The combination of these two nonverbal behaviors can more comprehensively describe the visualization of a depressive state. This proves that it is better to analyze multiple features in videos. This is also consistent with previous multimodal studies [39,46], and multichannel information is helpful for depression identification.

This study identified reliable and accurate predictive models. The psychological reliability and validity test method is applied to the depression recognition model validation, and more evaluation indicators are used to evaluate the validity of the model. Our model provides information about the most important variables to predict depressive state in primary healthcare. This method can be combined with patient interrogation and does not need other operations. Thus, it is simpler and more repeatable. This method can also be combined with monitoring technology for large-scale depression monitoring in some special occupations.

Our research also has some shortcomings. First, because we do not use the recorder to collect the subjects’ pure voice, we cannot incorporate voice into the establishment of a multimodal model. In fact, voice can provide much information for depression recognition. Future research may consider integrating audio-visual behaviors to predict a depressive state in primary healthcare, as a voice sample can be easily obtained without additional equipment or tasks. Second, when fusing multimodal features, we only simply splice the features instead of applying some more complex feature fusion methods, such as calculating the inner product. Complex feature fusion methods can bring more data, but the amount of data is also extremely large. Third, in future research, we will try to further verify the effectiveness of our method on the public dataset.

## Figures and Tables

**Figure 1 healthcare-10-02347-f001:**
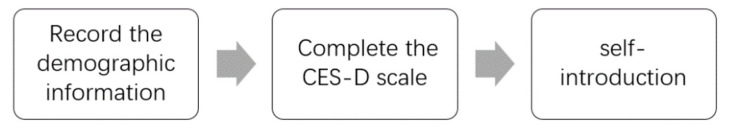
The overall process of the experiment.

**Figure 2 healthcare-10-02347-f002:**
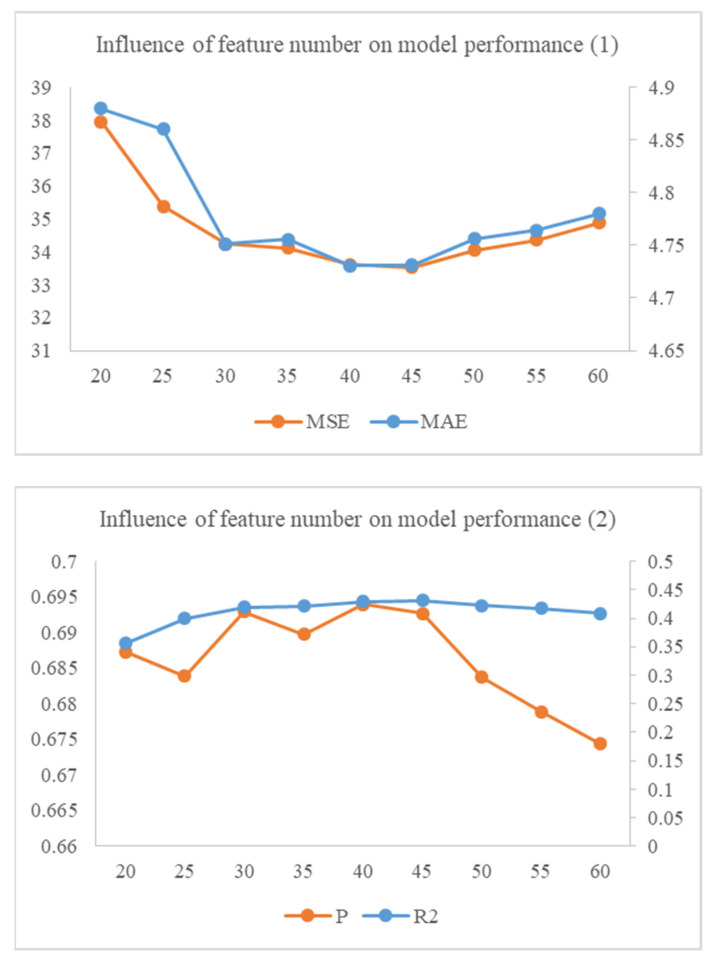
The trend of evaluation scores (r, MAE, MSE, R2) changing with the number of features. Smaller Mae and MSE scores represent that the predicted scores are closer to the real scores, while larger R2 and r values represent that the predicted scores are more related to the real scores.

**Figure 3 healthcare-10-02347-f003:**
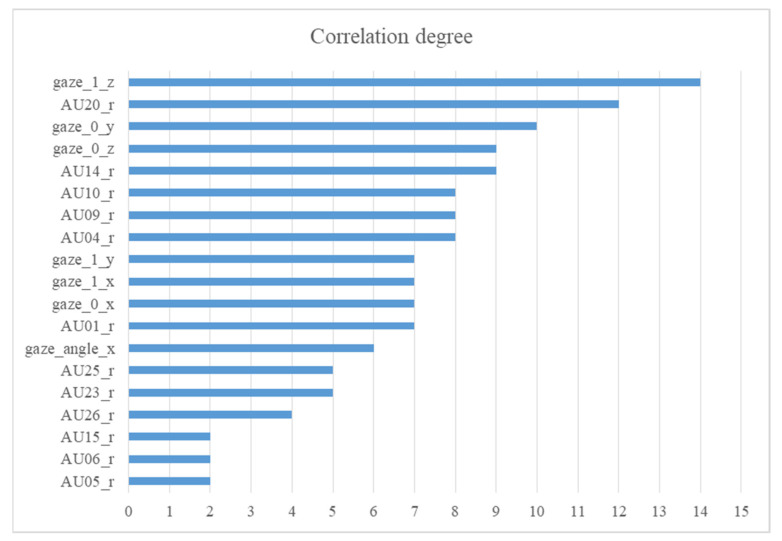
The ranking of the correlation degree between features and CES-D scores. A correlation degree is the sum of F values of selected principal components of a feature and CES-D score.

**Table 1 healthcare-10-02347-t001:** The specific meaning of visual features.

Visual Features	Index	Meaning
Facial activities	AU01_r	Inner Brow Raiser
	AU02_r	Outer Brow Raiser
	AU04_r	Brow Lowerer
	AU05_r	Upper Lid Raiser
	AU06_r	Cheek Raiser
	AU07_r	Lid Tightener
	AU09_r	Nose Wrinkler
	AU10_r	Upper Lip Raiser
	AU12_r	Lip Corner Puller
	AU14_r	Dimpler
	AU15_r	Lip Corner Depressor
	AU17_r	Chin Raiser
	AU20_r	Lip stretcher
	AU23_r	Lip Tightener
	AU25_r	Lips part
	AU26_r	Jaw Drop
	AU45_r	Blink
Eye_gaze	gaze_0_x	Eye gaze direction vector for the left eye in the y-axis direction
	gaze_0_y	Eye gaze direction vector for the left eye in the z-axis direction
	gaze_0_z	Eye gaze direction vector for the left eye in the x-axis direction
	gaze_1_x	Eye gaze direction vector for the right eye in the y-axis direction
	gaze_1_y	Eye gaze direction vector for the right eye in the z-axis direction
	gaze_1_z	Eye gaze direction vector for the right eye in the x-axis direction
	gaze_angle_x	Eye gaze direction in radians averaged for both eyes in the x-axis direction
	gaze_angle_y	Eye gaze direction in radians averaged for both eyes in the y-axis direction

AU represents action unit; x, y, and z represent three-dimensional coordinates, respectively.

**Table 2 healthcare-10-02347-t002:** Performance comparison of different models.

Algorithms	Feature Number	MAE	MSE	R2	r
KRR	40	5.413	40.948	0.305	0.628 ***
KRR-Periodic	40	4.731	**33.620**	**0.429**	**0.694 *****
SVR-linear	45	4.489	34.342	0.417	0.688 ***
SVR-Periodic	45	**4.488**	34.328	0.418	0.692 ***
SVR-RBF	40	4.736	40.403	0.315	0.601 ***

*** *p* < 0.001; Bold represents the best result.

**Table 3 healthcare-10-02347-t003:** Performance comparison of different characteristics.

Feature	Feature Number	MAE	MSE	R2	R
Facial	35	4.957	36.098	0.388	0.655 ***
Facial & gaze	40	**4.731**	**33.619**	**0.429**	**0.694 *****

*** *p* < 0.001.

## Data Availability

The datasets presented in this article are not readily available because raw data cannot be made public. If necessary, we can provide behavioral characteristic data. Requests to access the datasets should be directed to X.L.

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
