# Peer review of "Automatic Identification of a Depressive State in Primary Care"

_healthcare, 2022, doi:10.3390/healthcare10122347_

Round 1
Reviewer 1 Report
This paper estimates depression from gaze and facial expressions. However, the proposed method is a combination of past methods. These have already been discussed in many studies. Also, compared to the latest research, the estimation accuracy is low
Here are some comments.
If a model is to be proposed, it should be modeled and compared with the latest research.
These studies estimate depression from facial expressions alone, but this paper also suggests that facial expressions alone are an effective feature because the accuracy of facial expressions alone is higher than that of gaze + facial expressions.
Therefore, it is necessary to compare the accuracy of these studies with that of the present study.
Zhou, X., Jin, K., Shang, Y., & Guo, G. (2018). Visually interpretable representation learning for depression recognition from facial images. IEEE Transactions on Affective Computing, 11(3), 542-552.
Zhou, X., Wei, Z., Xu, M., Qu, S., & Guo, G. (2020). Facial depression recognition by deep joint label distribution and metric learning. IEEE Transactions on Affective Computing.
Haque, A., Guo, M., Miner, A. S., & Fei-Fei, L. (2018). Measuring depression symptom severity from spoken language and 3D facial expressions. arXiv preprint arXiv:1811.08592.
For model evaluation, a general depression dataset (AVEC2014) should be used for comparison.
Author Response
Thank you for your comments on my manuscript entitled " Automatic identification of a depressive state in primary care" which I submitted to Healthcare (Manuscript ID 1915971). I really appreciate your careful review. I am particularly grateful for your invitation for revision and resubmission and for the reviewers' recognition of the importance of new methods of depressive state recognition. Please see the attachment.

Reviewer 2 Report
This paper studies RAutomatic identification of a depressive state in primary care, it is important and interesting research. However, the innovation and contribution of the article still need to be highlighted. Is it innovation in experiment, algorithm or other aspects?
Author Response
Thank you for your comments on my manuscript entitled " Automatic identification of a depressive state in primary care" which I submitted to Healthcare (Manuscript ID 1915971). I really appreciate your careful review. I am particularly grateful for your invitation for revision and resubmission and for the reviewers' recognition of the importance of new methods of depressive state recognition.

Reviewer 3 Report
In this paper the authors aim to establish an automatic depression recognition model to be easily used in primary healthcare. The authors integrate facial activities and gaze data to establish an automatic recognition model of CES-D scores. It is assumed that the prediction model can predict CES-D scores effectively to indicate depressive states, and that the model fusing facial activities and gaze is better than the model containing only facial activities.
It is a well-structured paper, that needs some improvements and clarifications, as follows:
Abstract
- The authors mention about a Machine Learning algorithm that was used – KRR, but they do not provide any details regarding its abbreviations and an overview of its use.
- What algorithms were compared and under what criteria?
- You do not need to say twice the purpose of this study – the authors describe this purpose at the beginning of the abstract and at the end of the abstract. Please rephrase.
Introduction
- How are mild mental health problems are increased in severity? Please clarify this sentence
- Please provide more information regarding the first 26 references. You just provide the reference without indicating what these references and other studies provide. Putting just numbers is not sufficient.
- A paragraph is missing clearly describing the added value of this research, in comparison with other studies.
- A paragraph should be added indicating the structure of the rest of the document.
- You should also make a reference to other research studies making use of Machine Learning methodologies with different scopes and domains, including:
o Mavrogiorgou, A., Kiourtis, A., Kyriazis, D., & Themistocleous, M. (2017, September). A comparative study in data mining: clustering and classification capabilities. In European, Mediterranean, and Middle Eastern Conference on Information Systems (pp. 82-96). Springer, Cham.
o Khanam, J. J., & Foo, S. Y. (2021). A comparison of machine learning algorithms for diabetes prediction. ICT Express, 7(4), 432-439.
Method
- How did the participants were selected, randomly? Did they have to complete a consent form? What about the ethical and legal compliance actions that were considered regarding their data collection?
- The data acquisition experiment should be also depicted through a figure, indicating the overall process and steps
- What were the 20 questions that were included in the scale? Could you provide some examples?
- Why were the videos cut on 500 px x 500 px? Is it something standard?
Results
- You do not have to repeat the total number of participants
- What common algorithms were compared?
- I would like to see some more details regarding the comparative analysis that was performed, depicting the comparison criteria
Discussion
- Please provide more information regarding the limitations of your work, as well as the assumptions that were made
How do you plan to disseminate your results, and who are the stakeholders?
- What are your next steps and further enhancements?
Author Response

(The authors gave the same response as above.)

Round 2
Reviewer 1 Report
If you are proposing a new model, it is necessary to verify the model with a general-purpose dataset, or build a model that has been proposed in the past and conduct comparative verification.
I checked the revised paper.
The item I pointed out last time has not been corrected.
Author Response
I gratefully appreciate for your valuable comments. We try to search on the Internet and email the author, hoping to obtain the public data set t (AVEC2014). However, this dataset is still unavailable. So our method cannot be verified on the public dataset just now. In addition, previous studies cannot be compared temporarily because they do not have the same data set. However, we have supplemented the introduction of relevant literature, and explained in the discussion section that in the following research, we can further compare the effectiveness of our algorithm by replicating other research methods.
Reviewer 3 Report
All comments have been addressed
Author Response
Thank you for your comment and suggestion!